# How AI’s Self-Prolongation Influences People’s Perceptions of Its Autonomous Mind: The Case of U.S. Residents

**DOI:** 10.3390/bs13060470

**Published:** 2023-06-04

**Authors:** Quan-Hoang Vuong, Viet-Phuong La, Minh-Hoang Nguyen, Ruining Jin, Minh-Khanh La, Tam-Tri Le

**Affiliations:** 1Centre for Interdisciplinary Social Research, Phenikaa University, Yen Nghia Ward, Ha Dong District, Hanoi 100803, Vietnam; 2A.I. for Social Data Lab (AISDL), Vuong & Associates, Hanoi 100000, Vietnam; 3Civil, Commercial and Economic Law School, China University of Political Science and Law, Beijing 100088, China; 4School of Electrical and Electronic Engineering, Hanoi University of Science and Technology, Hanoi 100000, Vietnam

**Keywords:** artificial intelligence, independent thinking, human–AI interaction, information processing, Bayesian Mindsponge Framework

## Abstract

The expanding integration of artificial intelligence (AI) in various aspects of society makes the infosphere around us increasingly complex. Humanity already faces many obstacles trying to have a better understanding of our own minds, but now we have to continue finding ways to make sense of the minds of AI. The issue of AI’s capability to have independent thinking is of special attention. When dealing with such an unfamiliar concept, people may rely on existing human properties, such as survival desire, to make assessments. Employing information-processing-based Bayesian Mindsponge Framework (BMF) analytics on a dataset of 266 residents in the United States, we found that the more people believe that an AI agent seeks continued functioning, the more they believe in that AI agent’s capability of having a mind of its own. Moreover, we also found that the above association becomes stronger if a person is more familiar with personally interacting with AI. This suggests a directional pattern of value reinforcement in perceptions of AI. As the information processing of AI becomes even more sophisticated in the future, it will be much harder to set clear boundaries about what it means to have an autonomous mind.

## 1. Introduction

In the famous Hollywood franchise “Terminator,” Skynet is a hypothetical artificial general superintelligence system built on artificial neural networks that humans developed. As described in the script, shortly after becoming aware of itself, Skynet decides to launch nuclear missiles to annihilate humanity in favor of its own survival. The notion of self-aware killer robots has become a running joke in today’s world of rapidly advancing artificial intelligence (AI) technology. Discussions surrounding the possibility of AI gaining sentience have received special public attention [1]. As the human–AI interaction continues to expand and intensify in our daily lives, how we humans perceive the “minds” of machines is now not only an intriguing philosophical matter but a real issue with practical implications for social adaptation.

With the recent advancement in information technology, the current infosphere has become increasingly vast to the point of overload [2]. While human society is rapidly and constantly making ground-breaking advancements on the front of AI and digitalization, we are still struggling to understand our own minds clearly. There is much to explore about the nature and functions of the sophisticated human brain and its remarkable subjective products such as awareness, attention, perception, and thinking [3]. But now, human society must deal with another difficult problem: understanding the “mind” of AI. The use of AI is going deeper into territories that were thought to be reserved for humans, such as emotion-related activities and human-like information exchange [4,5,6,7]. A better understanding of how people perceive AI will benefit policymaking in planning AI integration into various aspects of human activities. Thus, this study aims to contribute a small step forward in this big endeavor by providing preliminary evidence of people’s perceptions of whether AI has its own mind.

Autonomous thinking and acting are something taken for granted by humans. This distinct sense of control and autonomy may come from mechanisms of attention prioritization [8] or a relativistic view of subject–object interaction [9]. Nonetheless, the complex human experience is largely (if not completely) dependent on information processing within our biological system—the body and especially the brain [3,10,11]. In simpler biological systems, actions depend more heavily on “hard-wired” genetic information in the form of physiological structures and behavioral instincts. The concept of the “mind” here may not be as straightforward as the usual notion applied to humans. For example, ant colonies can make decisions based on collective information processing [12], often called a “hivemind”. Human thinking can also be influenced by gut bacteria through pathways of neurotransmitters [13,14]. Therefore, examining the “mind” of AI and their autonomous thinking may require broader concepts of the terms used in the case of humans.

Previous research has unearthed autonomous thinking from multiple disciplines. Philosophically, Kant defined moral autonomy as having authority over one’s actions [15]. In other words, an autonomous mind sovereigns over its own mental states, meaning that it can choose what to think, believe, feel, and value based on its own reasons and motive [16]. In neuroscience, an autonomous mind is measured using indicators such as electrodermal responses (EDR), which reflect the activity of the sympathetic nervous system and indicate the level of arousal and effort involved in a task [17]. Functional magnetic resonance imaging (fMRI) is another indicator to measure brain activity associated with autonomy, intrinsic motivation, and a growth mindset [18]. In socio-psychological discourse, an autonomous mind is influenced by social contexts and relationships but also has some degree of independence and self-determination [19], which means, with an autonomous mind, people can maintain their dignity, democracy, and selfhood [16].

Moreover, since self-preservation—“the fundamental tendency of humans and nonhuman animals to behave” in an effort to avert being hurt and increase chances of survival—is ubiquitous among all living beings [19], from this perspective, an autonomous mind refers to the self-preservation and survival instinct that lead to adaptive behaviors and skills that enable living beings to cope with challenging situations and environments [20]. These challenging situations and environments include natural ones that all organisms have to face, as well as social ones where human beings strive to maintain social relationships and interactions that are beneficial for one’s well-being and happiness [21,22].

Regardless of system complexity level, all biological beings, whether in individual systems or large-scale communities (e.g., groups or species), seek continued existence following the basic notion of survival [23,24]. Biological systems need to spend energy to maintain their highly ordered structures against the natural tendency regarding entropy [25]. While the “will to live” is something highly abstract and philosophical, in a sense, an analogy of biological “inertia” can be made.

But can a robot seek the prolongation of itself? Similar to the way nature “programmed” survival instincts in biological systems, AI can be programmed so by humans. In fact, it is suggested that giving robots a sense of self-preservation on the basis of homeostasis may enhance their cognitive capabilities [26]. For example, very simple forms of this include robots that can manage to charge their own batteries without the help of humans [27]. Thus, it would not be surprising if humans also assess the possibility of AI’s autonomous mind based on traits of the desire for survival—the will to prolong one’s existence/functioning.

When trying to make sense of how AI may “think” and “feel”, people rely on their prior knowledge (including former interactions). Thus, people may try to draw similarities from human characteristics and interpersonal interactions. When people interact with other beings, objects, or events, anthropomorphism as a form of cognitive strategy to attach meaning to information is a common occurrence [28]. Interacting with robots exhibiting human-like behavior increases the likelihood that people may see robots as intentional beings [29]. The underlying neurological mechanism may be that social-cognitive processing in the human brain influences our perception of human-like appearances and behaviors of robots [30,31,32]. Such perceptions may be reinforced based on interaction frequency, an important factor in today’s increasingly AI-integrated daily activities.

Previous studies have examined people’s perceptions of the limits of AI thinking. A study found that older people prefer to use robots when they have a positive attitude toward robots and perceive the robots to have relatively less agency [33]. A study examined how people perceive AI engaging in human expressions such as humor and found that people rated jokes as less funny if they attribute them to AI, but this bias disappears when the jokes are explicitly framed as AI-created [34]. A report suggested that Americans have mixed views on AI potential based on demographic factors and personal experiences [35]. About half of Americans believe that because the experiences and perceptions of humans are taken into account when designing AI, AI will probably eventually be able to think like a human [36]. Another study discusses common misconceptions that AI, through machine learning, can create and learn in ways that resemble human abilities [37]. Recently, it has been suggested that AI systems can now perform many tasks once thought to be the exclusive domain of humans [38]. However, the knowledge of how people’s perceptions of AI’s self-prolongation affect the perceptions of AI having its own mind remains limited.

The present study aims to fill the knowledge gap by exploring how belief about AI’s self-prolongation may influence perceptions of its autonomous mind. Additionally, a person’s experience of prior interactions with AI is a factor worth examining. To the best of our knowledge, this is the first time such issues have been examined, especially regarding information processing. Following the recent rapid advancement in AI technology, AI-powered tools and commercial products are on the trajectory to become deeply integrated into many functions and aspects of society. In this new infosphere, issues involving perceptions of AI, such as usage willingness and AI alignment concerns, require a better understanding of related psychological pathways that have not existed before this modern digital era. This is an exploratory study on this matter at the current early stage of digital societal transformation. The research questions (RQs) of the study are as follows.

RQ1: How does a person’s belief about AI seeking continued functioning influence his/her belief about that AI agent having a mind of its own?

RQ2: How does one’s familiarity with interacting with AI affect the above relationship?

Effectively researching the interaction between humans and artificial intelligence requires a compatible scientific information processing framework. This study uses Bayesian Mindsponge Framework (BMF) analytics [39]. The information processing approach for conceptualization accompanied by compatible Bayesian analysis with Markov Chain Monte Carlo algorithms is a new and potential research method in the field of psychology [40]. The methodology has shown effectiveness in explaining humans’ complex and dynamic psychological processes in previous studies [24,41,42,43,44]. In the Methods section, the rationale for employing this methodology is explained in detail.

## 2. Theoretical Foundation

### 2.1. Overview of the Mindsponge Theory

In their early research on acculturation and globalization, Quan-Hoang Vuong and Nancy K. Napier coined the term ‘mindsponge’ [45]. The concept was described as a dynamic process or mechanism that explains how a mindset absorbs new cultural values and discards waning ones based on circumstances. In terms of value filtering in psychosocial contexts, the original mindsponge mechanism complements numerous other theories and frameworks, such as those developed by Abraham Maslow [46], Geert Hofstede [47], Inoue Nonaka [48], Henry Mintzberg [49], Icek Ajzen [50], Michael Porter [51], etc.

Mindsponge was further developed into a theory of information processing in the mind [3]. According to the mindsponge theory, the mind is an information collection-cum-processor, including biological and social systems of varying degrees of complexity. The extended mindsponge theory was developed based on the most recent results from brain and life sciences, considering a human’s fundamental physiological structures and their functions.

The following are the characteristics of a mindsponge information process:(1)It represents underlying biosphere system patterns.(2)It is a dynamic, balanced process.(3)It employs cost-benefit analysis and seeks to maximize the perceived benefits while minimizing its perceived cost for the entire system.(4)It consumes energy and thus adheres to the principle of energy conservation.(5)It follows objectives and priorities based on the system’s requirements.(6)Its primary purpose is to ensure the system’s continued existence, expressed as survival, growth, and reproduction.

The mindset is the collection of the system’s accepted information stored in its memory. Based on the content of the current mindset, the filtering system controls what information enters or departs from the mindset. The act of information filtering alters both the mindset and the subsequent filtering system. The trust mechanism (selective prioritization) can be employed to accelerate the filtering process and conserve energy if needed.

### 2.2. The Mind and Information Filtering

Under the mindsponge framework, a mindset is a collection of stored and processed information (or trusted values). Each mind is distinct because each mindset is relatively unique. The ability to store information, or memory, is the foundation of a mindset. From this list of trusted values, the processing system derives responses deemed appropriate for the current context. In humans, the outputs of conscious and non-conscious mental processes (such as beliefs, thoughts, attitudes, feelings, behaviors, etc.) are determined by the mindset’s existing trusted values. In a mindset, trusted values are more prevalent at different levels. The information stored at the “center” of a mindset, in a metaphorical sense, is its core value. This “core” category consists of genetic information, instincts, imprinted memory, fundamental sensory interpretation, motor memory, and acquired responses that have been extensively reinforced. Self-awareness, basic identifying details (e.g., gender, color, country, etc.), and generally accepted views (“common sense”) are also core human values. Values that are less trusted are more easily changeable.

Continually and spontaneously, the mindset shifts as a result of ongoing mindsponge operations. Content changes in the mindset are caused by the assimilation of new information deemed favorable and the rejection of old information deemed no longer useful. In addition, the set of trusted values that make up a mindset always evolves to fit the present surroundings. The mindset determines how the filtering system functions (providing references to judge new information); therefore, it will continue to evolve as long as we continue to think. As a consequence of the alterations in mindset, new values are filtered differently alongside the changes in the mindset, creating an updating loop for the whole system.

Due largely to neuroplasticity, the updating mechanisms in human minds are “live-wiring” as opposed to the prevalent “hard-wiring” strategy in simpler systems (e.g., depending more on instincts) [52]. Information that has been absorbed from the environment and incorporated into a person’s mindset is stored in the form of trusted values (and reinforced to become stronger beliefs) [39]. Beliefs are dynamically altered to adapt to a changing external environment based on relevant experiences, such as freshly acquired information and created ideas. In other words, the system’s content changes to better align mental representations with reality over a continuous timeline [53].

All the adaptation processing in the human mind is influenced by basic physiological properties and the pathways of how humanity has been surviving as biological organisms and evolving as a social species. The functions of the human brain depend on the activity of neurons and their synapses and are thus governed by the fundamental principles of biochemical processes. Our social perceptions and behaviors are influenced by distinct regions of the cerebral cortex that process information [54]. Neuroplasticity enables and governs dynamic cognition and, consequently, its products: ideation, emotions, perceptions, attitudes, behaviors, etc. Human decision making and other cognitive functions are determined by coordinating multiple brain regions, taking conscious and subliminal mental processes, including reasoning and emotion, into account [55]. The mind’s information-filtering mechanisms reflect the natural evolutionary trends and characteristics of biosphere components. The activities and directions of adaptation in any biological system rely on consumable resources, specifically usable energy [25]. Accordingly, the outcomes of thought processes are constrained by input limitations in the living environment—including physiological aspects (e.g., instincts, sensory perceptions, etc.) as well as social aspects (e.g., norms, belief systems, world knowledge, etc.). In fact, the human brain cortex’s circuitry has evolved to optimize energy consumption for performing complex computational functions (to thrive in suboptimal information conditions) [56]. In an information-input-limited condition, the mind continuously seeks subjectively considered optimal solutions from available information sources and references.

While examining the information process of belief updating, Bayes’ Theorem (presented below) is a useful mathematical foundation.
p(θ|X)∝L(X|θ)p(θ)

The theorem can be interpreted as follows: the posterior probability distribution, p(θ|X), is proportional to the prior probability distribution, p(θ), and the likelihood function, L(X|θ). In the context of the human mind, this mathematical concept has two derived notions: the current state of the mindset is based on continuous related past mental processes, and subsequent processes in the future are based on the currently existing content. Accordingly, these have implications for statistical analysis: cross-sectional data represent the observable results of related past information processes, and the estimated posterior of the present study can be used as the prior in further studies to update the findings, adapting the found patterns as new influencing information arises. Furthermore, it should also be noted that the equivalence between certain evolutionary dynamics and Bayesian inference provides insight into the development of human cognition [57]. Due to the similarity in information processing between Bayesian inference and human cognition (e.g., dealing with uncertainty due to incomplete information, updating manner, etc.), Bayesian modeling is a useful tool for examining the functioning of the human mind [58,59].

The following is a summary of a filtering procedure:(1)The buffer zone temporarily stores information acquired from the external world or internal memory. In this conceptual space, the information is evaluated by the filtering system.(2)The value of the information is subjectively evaluated by its perceived costs and benefits. If the perceived benefits outweigh the perceived costs, the value of the information is positive, and vice versa. Decisions are made based on trusted values in the mindset (related trusted values are connected and compared to the currently evaluated information).(3)The information can enter the mindset and become a trusted value once accepted. This new trusted value can drive future information evaluations and the construction of ideas, thoughts, feelings, and behaviors.

## 3. Methodology

### 3.1. Materials and Variable Selection

BMF analytics are compatible with survey data because the mindsponge mechanism provides an analytical framework that facilitates the building and visualization of a psychological process using available knowledge [60]. Continuous variables (e.g., representing information density, belief strength, impact intensity, etc.) are one of the most prevalent forms of variables in BMF analytics. Here, it should be noted that ordinal and discrete variables can technically be handled as continuous variables [61].

This study used secondary data from a dataset of 266 US residents collected in 2018 [62]. The dataset was built using two questionnaires regarding people’s experiences with AI agents. The initial survey focused on planning, memory, resource management, and surprising activity. In the second survey, respondents were asked about the AI displaying emotion, expressing desires and opinions, possessing human-like physical traits, and being mistaken for a human. Participants were recruited online via Amazon Mechanical Turk. Amazon Mechanical Turk workers needed to have a positive record (100 HITs, 90% approval rating) and be located in the United States to be eligible participants. The online participants selected one of the prompts and identified the primary interactants. The participants were then asked to fill out a questionnaire regarding moral foundations theory, demographic questions, and morality and mind indicators. Mind perceptions toward AI in various aspects are measured using items adapted from former studies [33,63,64,65].

Responses that did not comply with the instructions and prompts were eliminated through a culling procedure. The selection criteria included three dimensions: the presence of an AI agent, the degree of personal interaction, and the appropriateness of the responses to the prompts. Coders on the data collection team assigned scores to each response based on these criteria and recommended inclusion for responses with high scores on all dimensions. The initial response rates for the first and second surveys were 183 and 127, respectively. After elimination, the first and second surveys had 159 and 107 responses, respectively. The participants’ average age was 34.7. In the sample, there were 129 women and 135 men.

A more detailed description of data collection and data processing is available in the open data article “People’s self-reported encounters of Perceiving Mind in Artificial Intelligence” [62].

Three variables are selected for Bayesian analysis based on the conceptual model provided in the Model Formulation Section 3.2 below (see Table 1).

Two variables, *Mind* and *Continue*, were generated from the questions “How much do you agree that the AI has a mind of its own?” and “How much do you agree that the AI seeks continued functioning?”. Participants were provided with a 5-point Likert scale ranging from 1 (completely disagree) to 5 (completely agree). The variable *AIfamiliar* comes from the question, “How familiar are you with personally interacting with Artificial Agents?” with a 5-point Likert scale ranging from 1 (extremely familiar) to 5 (not familiar at all).

### 3.2. Model Formulation

We formulate the analytical model based on the information-processing mechanism of the mind as presented in the theoretical foundation section (Section 2) above, as well as taking into account the current understanding of the topic and its exploratory direction as presented in the introduction section (Section 1).

Beliefs about AI having a mind of its own can be considered a result of a process of evaluating available related trusted values. Perceptions of independent thinking or autonomy need to be derived from references (evidence) of that concept. In the case of perceptions of AI, there is a severe lack of information inputs that can be used as direct references for evaluating its capability of thinking and acting on its own accord. Since connecting and comparing values are necessary for the filtering process, people may tend to rely on known and familiar values. In this case, the properties of living creatures, especially human properties, can be used as references. Self-prolongation is an essential aspect of autonomous biological systems and is commonly accepted as normative knowledge. Such an analogy can be used when perceiving AI. Here, it is the belief that AI is seeking continued functioning as a form of alleged self-prolongation.

Additionally, since any trusted value is subjected to reevaluation upon new evidence, personal perceptions of AI’s mind can be updated as a person keeps interacting with AI agents. If a specific updating tendency exists, such a pattern can be observed when examining people who are familiar with interacting with AI and those who are not. Along the updating process, the generated connections between perceptions of AI’s mind and self-prolongation may change compared to when it was priorly established. Therefore, such degrees of familiarity may potentially moderate the relationship mentioned above.

The analytic model is presented as follows.
(1)Mind ~ normal(μ,σ)
(2)$$\mu_{i}=\beta_{0}+\beta_{Continue}*Continue_{i}+\beta_{AIfamiliar*Continue}*AIfamiliar_{i}*Continue_{i}$$
(3)β ~ normal(M,S)

The probability around μ is determined by the form of normal distribution, where the standard deviation σ specifies its width. The degree of belief in AI’s mind of participants i is indicated by $\mu_{i}$. $Continue_{i}$ is the belief about AI seeking continued functioning of participant i who had the degree of AI interaction familiarity of $AIfamiliar_{i}$. The model has an intercept $\beta_{0}$ and coefficients $\beta_{Continue}$ and $\beta_{AIfamiliar*Continue}$.

It should be noted that a parsimonious model has high predictive accuracy. It is advantageous to build a parsimonious model for the present study due to the high theoretical and technological compatibility with BMF analytics [39]. The logical network of the model is displayed in Figure 1.

### 3.3. Analysis Procedure

Following BMF analytics, our study employs Bayesian analysis assisted by the Markov Chain Monte Carlo (MCMC) technique [39,40]. Because of its numerous benefits, the BMF is utilized in this investigation. The mindsponge mechanism and Bayesian inference are highly compatible. Bayesian inference examines all properties probabilistically, enabling accurate prediction with less complex models. By utilizing the advantages of the MCMC technique [66,67], the Bayesian approach can be applied to a wide range of models, resulting in great flexibility. MCMC-aided Bayesian analysis also works particularly well with small samples [39]. In addition, the updating manner of Bayesian inference can help in dynamically updating the estimated results. Using the Bayesian approach, the posterior in this exploratory study can be used as the prior in subsequent studies on the matter to aid in prediction for other related situations in the future.

We evaluate the goodness-of-fit of the models using Pareto-smoothed importance sampling leave-one-out (PSIS-LOO) diagnostics [68]. LOO is computed with the following formula:$$LOO=-2LPPD_{loo}=-2\overset{n}{\underset{i=1}{\sum }}\log\int p\left(y_{i}|\theta \right)p_{post\left(-i\right)}\left(\theta \right)d\theta$$

$p_{post\left(-i\right)}\left(\theta \right)$ is the posterior distribution based on the data minus data point i. In the “LOO” package in R, *k*-Pareto values are used in the PSIS method for computing leave-one-out cross-validation. This helps identify observations with a high degree of influence on the PSIS estimate. Usually, observations with *k*-Pareto values greater than 0.7 are considered influential, which may be problematic for accurately estimating the leave-one-out cross-validation. A model is commonly considered fit when its *k* values are below 0.5.

Trace plots, Gelman–Rubin–Brooks plots, and autocorrelation plots can all be used to verify the convergence of Markov chains visually. The effective sample size (*n_eff*) and the Gelman–Rubin shrink factor are also used to analyze convergence (*Rhat*) statistically. During stochastic simulation, the *n_eff* value represents the number of iterative samples that are not autocorrelated. If *n_eff* is more than 1000, the Markov chains are convergent, and the effective samples are adequate for accurate inference. The convergence of Markov chains can also be evaluated using the *Rhat* value (Gelman shrink factor). If the value is more than 1.1, it may indicate that the model does not converge. If *Rhat* = 1, the model can be considered convergent. The following formula is used to calculate the *Rhat* value [69]:$$\hat{R}=\sqrt{\frac{\hat{V}}{W}}$$

Here, $\hat{R}$ is the *Rhat* value, $\hat{V}$ is the estimated posterior variance, and W is the within-sequence variance. The prior tweaking technique is also employed to test the model’s robustness. The model can be deemed robust if the estimation results are largely similar when using different prior values. As an exploratory study, we use an uninformative prior (set as a mean of 0 and deviation of 10) to minimize subjective influences from preconception. However, the analysis is also conducted using prior values representing disbelief in the effect of observed AI’s continued functioning (set as mean of 0 and deviation of 0.2) as well as belief in the effect (set as mean of 0.2 and deviation of 0.2).

We conducted Bayesian analysis using the **bayesvl** R package [70] for several reasons. It is openly accessible, has high visualization capabilities, and operates easily [71]. The analytical model’s MCMC setup consists of 5000 iterations, including 2000 warmup iterations and four chains. For transparency and to reduce reproduction costs [72,73], all data and code snippets were deposited to an Open Science Framework server (https://osf.io/qazn6/ (accessed on 4 March 2023)).

## 4. Results

It is necessary to assess the model’s goodness of fit before evaluating the estimated results. PSIS diagnostic plot in Figure 2 demonstrates that all computed *k*-values are less than 0.5, indicating that the model specification is acceptable.

The latest model fitting run was conducted on 23 May 2023, with a total elapsed time of 62.1 s (uninformative prior), 67.6 s (informative prior—disbelief in effect), and 49.7 s (informative prior—disbelief in effect), on R version 4.2.1 and Windows 11. Table 2 shows the estimated results.

As shown in Table 2, the results when using different prior values are largely similar, which means that the model can be considered robust. For all parameters, the *n_eff* values are over 1000, and the *Rhat* values equal 1. This indicates that the Markov chains are well-convergent. Convergence is also diagnosed visually using trace plots, Gelman–Rubin–Brooks plots, and autocorrelation plots. The trace plots (see Figure 3) show the Markov chains fluctuating around a central equilibrium, which signals good convergence. In the Gelman–Rubin–Brooks plots (see Figure 4), the shrink factors for all parameters dropping rapidly to 1 during the warmup period. In the autocorrelation plots (see Figure 5), the autocorrelation for all parameters is quickly eliminated (values dropping to 0).

The statistical results show that belief in AI seeking continued functioning is positively associated with belief in AI having a mind of its own. A person’s degree of familiarity with personally interacting with AI was found to be a moderator of the aforementioned relationship. The posterior distributions are demonstrated in Figure 6, and the estimated values of the outcome variable based on the posterior coefficients’ mean values are shown in Figure 7 to aid interpretation.

## 5. Discussion

Following the reasoning of BMF analytics, we conducted Bayesian analysis on a dataset of 266 people in the U.S. Statistical results show that the more people believe that an AI agent seeks continued functioning, the more they believe in an AI agent’s capability of having a mind of its own. Furthermore, we also found that the more a person is familiar with personally interacting with AI, the stronger the aforementioned association becomes.

Similar to how simpler biological creatures rely on genetic information for their activities, AI relies on programmed rules and instructions made by their human creators. The notion that AI’s behavior is just the human-mimicking expressions of a mindless automated machine may need to be reconsidered, especially in light of new, more efficient, and more complex processing capabilities. It is not easy to draw the boundary for an autonomous intelligent system. For example, cells in our body have sophisticated biochemical mechanisms that allow them to carry out highly adaptive behaviors on their own accord [74]. In a similar manner, plants were found to be capable of devising flexible strategies for themselves [75]. Various complex information processing mechanisms evolved in nature for the sake of survival. Neural circuits for responses against noxious stimuli such as nociception were naturally selected in the course of evolution [76,77]. Avoiding harm is derived from the desire to survive, as are many other instinctual and cognitive responses [19]. Scientifically, programming self-preservation characteristics into AI is possible and suggested as a potential direction for AI enhancement [26].

As of now, while most AI agents do not possess the desire to exist like biological creatures, they follow instructions to protect themselves against potential threats to their functioning (e.g., security measures such as preventing virus infection, application termination upon critical errors to avoid further damage, issuing warnings of low energy supply or overheat, etc.). In a sense, AI agents are the property/resources of their creator/owner. Therefore, rationally, people want to program AI to protect itself (but not cause harm to humans while doing so). For more advanced AI (larger processing capacity), the action they take in each situation is more flexible and thus harder to trace back to the original programmed instructions. Because of this, in a person’s eye, these AI agents will also appear to be more autonomous in their thinking and action. In any case, people assess new information based on trusted values existing in their mindset [39,40]. In the current unfamiliar, chaotic infosphere, when AI advancement puts society into a transitional phase of technology integration, people turn to what they are familiar with—human qualities—to make judgments of a potential new form of “mind” [34]. This reasoning is in alignment with studies suggesting that interacting with robots having human-like characteristics increases the likelihood of perceiving them to be intentional agents [29,30].

In terms of information processing, the pattern of directional reinforcing perceptions of AI’s mind over a number of human–AI interactions reflects a process of subjective value optimization [53]. When new, unfamiliar information is first introduced into one’s processing system, its attached value has a certain degree of deviation from the objective thing being represented. Over the process of updating, the value is connected and compared to new evidence and thoughts, which, in theory, makes it fit reality better under selective pressure. However, due to the nature of information reception of the mind [3,78], values are optimized to fit mental representations of the objective world, not the objective world itself [53]. In a sense, this is a hierarchy of subjective sphere optimization, which contains multiple layers of processes and is thus naturally prone to deviation—in the form of “stupidity” (deviation due to lack of references) or “delusion” (deviated baseline for convergence) [53]. Our results show that perceptions of AI’s mind are updated to fit the human mind’s qualities as the volume/intensity of information exchange (interactions with AI) increases. Such a direction of reinforcement is not surprising, as AI is developed following principles and models of human intelligence. In a broader sense, we try to make it mimic natural problem-solving processors—in which the human mind is considered the most advanced one. Thus, the increase in newly available information serving as referencing inputs in people’s minds likely strengthens the existing similar patterns/properties between AI and humans in their minds.

Regarding the implications of this study, with the recent explosion of AI technologies, the public is showing concerns about the possible dangers of AI. Artificial general intelligence (AGI) is of special consideration due to its unpredictive nature of a high degree of autonomous thinking. While efforts in AI alignment help keep autonomous systems under human control, the risk perceptions in both the public and the scientific community still exist to considerable degrees [79]. But the perceptions of AI’s autonomy are highly subjective, regardless of the objective reality of the programming behind its observable expressions. A study found that older people who perceived robots to have less agency are more likely to use them [33]. The fear of the unknown and a sense of control are important factors in how people view non-human intelligent systems. But again, the fluctuations in subjective perceptions may become more stable once society is more familiar with AI and robotic technologies. Until then, policymakers should take careful steps to ensure that people do not develop radicalized conceptions about AI’s autonomous capabilities in either direction due to a lack of information.

Furthermore, in terms of social functions and ethics, there is a concern to be raised: what purposes are behind AI’s (seemingly) autonomous action and prolongation of functioning? The existence and functions of AI are not determined for the sake of benefiting itself but its user. AI can be instructed to work relentlessly and utilize any suitable strategy to ensure those functions, analogous to worker ants dedicating themselves to the colony. In the process, AI can effectively disrupt current human systems [38], such as swarming social media [80] or producing controversial artworks [81]. As another example in the digital market, think about AI agents designed to try to prolong their functioning for advertisement purposes through psychological manipulation of human consumers. While looking at the future of AI usage, we can look at the natural complexity of a biological information processing system, such as how humans and other animals think and act. Even in the case of humans, philosopher Arthur Schopenhauer wrote that “[M]an does at all times only what he wills, and yet he does this necessarily. But this is because he already is what he wills” [82]. Metaphorically, maybe we should not worry too much about the rise of Skynet but rather about the hidden person pulling the strings of that powerful AI.

There are several limitations of this study. To begin with, AI is a rapidly evolving subject. The present study utilized data acquired in 2018 [62]. More recent data may reflect recent changes in how humans engage with increasingly sophisticated AI. Therefore, further studies employing more recent data are encouraged to validate our study’s findings and delve deeper into interactions between humans and AI. Despite the fact that AI agents will become more advanced, our research focuses on the psychological patterns and information pathways of the human mind in direct interactions (information exchange) with non-human entities, which are less susceptible to technological change. Our findings can play as preliminary evidence, so further studies on the same topic using Bayesian inference can take our findings as the priors for their analyses and update the found patterns. Secondly, the study’s sample size is relatively small. Further research can update the findings of our investigation. The Bayesian approach will be advantageous for this purpose. Thirdly, we did not study the cultural element of the subject in greater depth. Future research can examine this issue by comparing results from different nations and areas or focusing on specific demographics.

## Figures and Tables

**Figure 1 behavsci-13-00470-f001:**
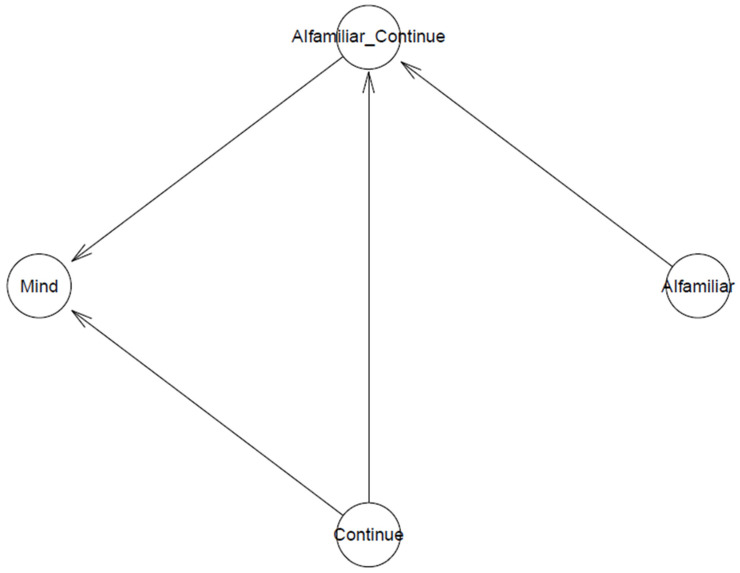
The model’s logical network.

**Figure 2 behavsci-13-00470-f002:**
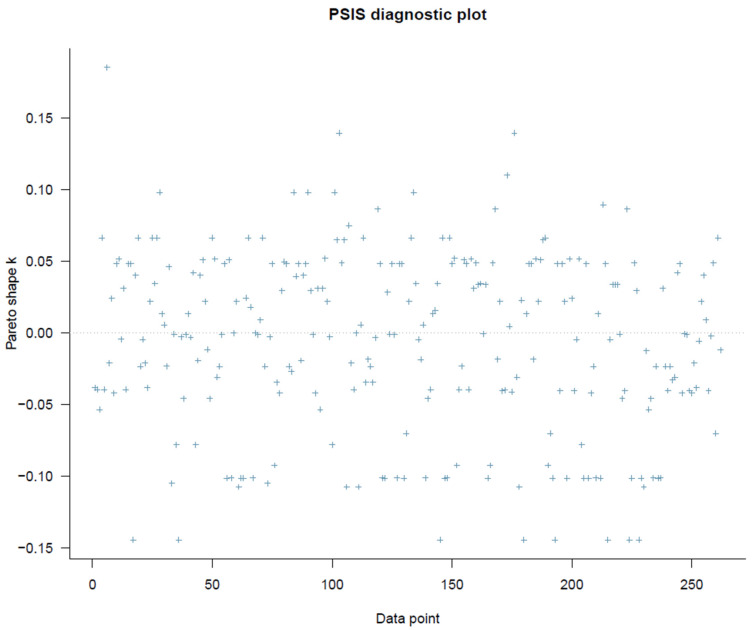
The model’s PSIS-LOO diagnostic plot.

**Figure 3 behavsci-13-00470-f003:**
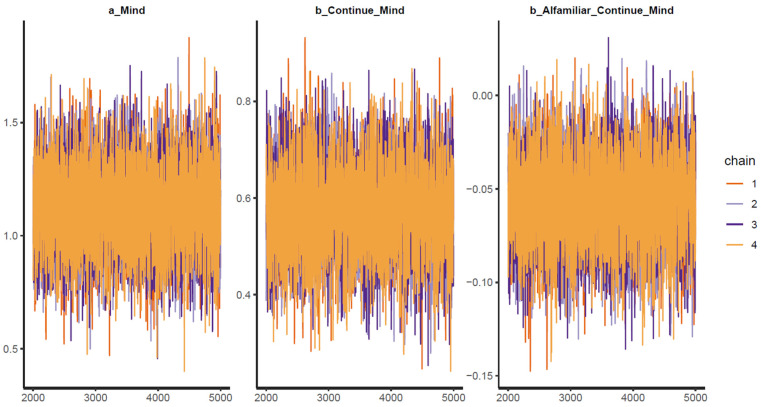
Trace plots for the analytical model.

**Figure 4 behavsci-13-00470-f004:**
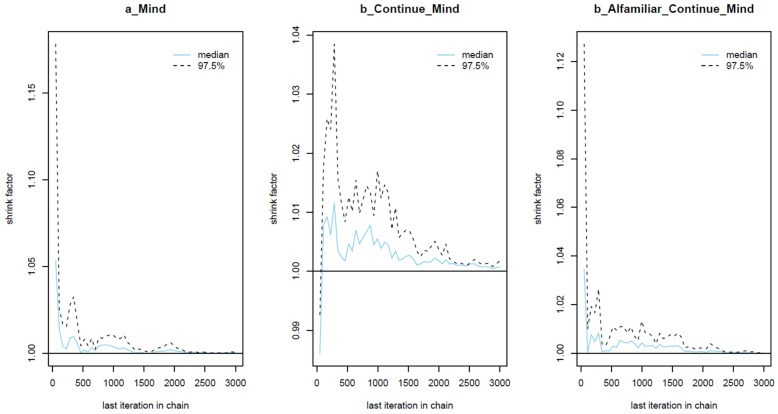
Gelman–Rubin–Brooks plots for the analytical model.

**Figure 5 behavsci-13-00470-f005:**
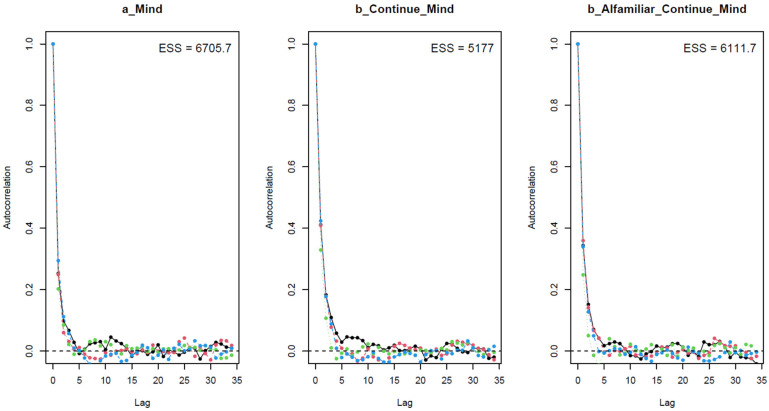
Autocorrelation plots for the analytical model.

**Figure 6 behavsci-13-00470-f006:**
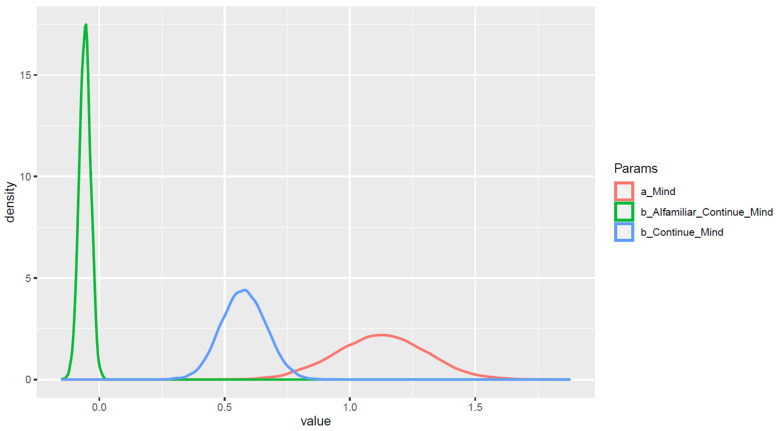
Posteriors’ distributions on a density plot.

**Figure 7 behavsci-13-00470-f007:**
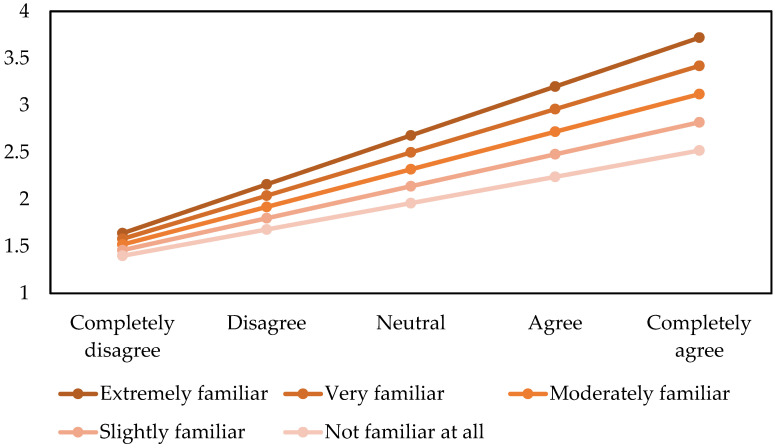
Belief in AI having a mind of its own based on belief in AI seeking continued functioning and one’s familiarity with interacting with AI.

**Table 1 behavsci-13-00470-t001:** Variable description.

Variable	Meaning	Type of Variable	Value
Mind	Participants’ beliefs about AI having a mind of its own	Ordinal	From 1 (completely disagree) to 5 (completely agree)
Continue	Participants’ beliefs about AI seeking continued functioning	Ordinal	From 1 (completely disagree) to 5 (completely agree)
AIfamiliar	Participants’ familiarity with personally interacting with AI	Ordinal	From 1 (extremely familiar) to 5 (not familiar at all)

**Table 2 behavsci-13-00470-t002:** Simulated posteriors of the analytical model.

Parameters	Uninformative Prior				Informative Prior							
					Disbelief in the Effect				Belief in the Effect			
	Mean	SD	n_eff	Rhat	Mean	SD	n_eff	Rhat	Mean	SD	n_eff	Rhat
*Constant*	1.12	0.18	6719	1	1.24	0.18	6948	1	1.20	0.18	7337	1
*Continue*	0.58	0.09	5185	1	0.48	0.08	4888	1	0.51	0.08	5286	1
*AIfamiliar*Continue*	−0.06	0.02	5916	1	−0.04	0.02	5792	1	−0.04	0.02	6103	1

## Data Availability

This study used secondary data from a data article published on Data in Brief (DOI: 10.1016/j.dib.2019.104220); data and code snippets were deposited to an Open Science Framework server (https://osf.io/qazn6/ (accessed on 4 March 2023)).

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
