# Peer review of "How AI’s Self-Prolongation Influences People’s Perceptions of Its Autonomous Mind: The Case of U.S. Residents"

_behavsci, 2023, doi:10.3390/bs13060470_

Round 1

Reviewer 1 Report

In this study, the authors conducted Bayesian analysis on a dataset of 266 people in the U.S. and found that the more people believe that an AI agent seeks continued functioning, the more they believe in that AI agent’s capability of having a mind of its own. Additionally, the more a person is familiar with personally interacting with AI, the stronger the association becomes. This suggests a directional pattern of value reinforcement in perceptions of AI. The authors posit that as the information processing of AI becomes more sophisticated in the future, it will be harder to set clear boundaries about what it means to have an autonomous mind.

The article requires a thorough analysis to engage the state of the art, justify the use of the methods, and explain why it is thought to be a research problem worth solving. In addition, it is necessary to provide information on the "details about how the instrument/questionnaire was developed," "who the participants were," and "the selection criteria of participants."

My observations are listed below:

1.    The scientific issue with the existing evaluation is not addressed at all in the introduction. It needs to be explained clearly and point by point. In fact, the authors have not conducted a thorough literature review, nor they have performed an analysis of the applicable theories in order to comprehend the state of the art and to support their methodology. 

2.    It is necessary to provide information on the "details about how the instrument/questionnaire was developed," "who the participants of this study were," and "the what the selection criteria of the participants."

3.    It is advised to describe the research purpose and objectives of this study, as well as what the authors have done to address the stated research problem, at the end of the introduction.

4.    A section at the end should mention other works and study limitations.

5.    At the end of the introduction, please specifically describe your contributions in comparisons with the latest related works. Please provide a comparison table that list related works and say how your work stands out amongst those. Comparison should be between related works with some differences. The experiments should be broadened to include more analysis, comparisons with other indices/baselines, and comparisons between your suggested method and recently developed state-of-the-art techniques.

6.    Therefore, I advise you to investigate more unresolved research questions in this field and to add at least 8-12 new ones.

7.    The results are presented as scattered or inconsistent graphs. They should be compared with the results of other state-of-the-art publications.

8.    Reference format must be uniform.

9.    The authors should proofread the entire document again for grammatical and typographical problems.

Author Response

Dear Reviewer,
Thank you for reviewing our manuscript “How AI’s self-prolongation influences People’s Perceptions of its autonomous mind: The Case of U.S. Residents” and providing valuable suggestions. We appreciate your feedback and have revised and improved our manuscript accordingly. We have attached the revised manuscript and a detailed response letter, explaining how we addressed each suggestion. We hope that you will be satisfied with our revisions and accept our manuscript. If you have any further questions or suggestions, please feel free to contact us. Thank you again for your time and effort.
Sincerely,
Ruining Jin

Reviewer 2 Report

You said Bayes’ Theorem (presented below) is a useful mathematical foundation for information process of belief updating, but it lacks justification. For example, why a moderation/mediation analysis in a regression model wouldn't suffice? Also, the equation was not well explained in the context.

It is not clear how the three variables were selected amongst all variables in the two questionnaires for Bayesian analysis. 

The section Model Formulation lacks theoretical foundation and evidence from literature.

You said that the Bayesian approach reduced the risk of over-dependence on the p-value, but equally it has the risk of overfitting which should have been discussed. Ideally two approaches could be compared in the present study and/or using evidence from literature.

Discussions lack implications and recommendations for future research and practice. Regarding the research question per se, the motivation/value of studying perception of AI having its own mind does not sound strong or persuasive - this perception is still very subjective and metaphorical given the fact that AI (including its self-learning/prolongation) is based on programming and pretty much predictable. Overall, a stronger rationale of both the topic and methodology needs to be provided.

Author Response

(The authors gave the same response as above.)

Reviewer 3 Report

Using data from a previous study, the authors seek to answer the following questions:

RQ1: How does a person’s belief about AI seeking continued functioning influence his/her belief about that AI agent having a mind of its own?

RQ2: How does one’s familiarity with interacting with AI affect the above relationship?

The author's technical approach is sound and their results show strong correlations.  This study is quite interesting and should be furthered with other studies to help determine levels of engagement and understanding of interactions with AI agents.

Author Response

(The authors gave the same response as above.)

Reviewer 4 Report

The manuscript has used two types of citations. E.g [1] and Vuong, La, et al., 2022).

Also the references listing is not according to the mdpi's format.

The paper lacks related research and a literature review.

The Methodology must be a separate section and before that describe all the theoretical concepts or other models and materials used in a separate Section e.g., the Background section.

The authors are formulating research questions in 2023 while utilizing the dataset collected in 2019. The authors are formulating research questions in 2023 while utilizing the dataset collected in 2019. Significant advancements in AI have occurred in recent years, which may have influenced people’s perceptions of AI and its “mind”.  As AI is evolving, it is, therefore, natural for people’s understanding and perspectives to evolve as well. So it would be better to report this manuscript with recent data collected from individuals.  

Author Response

(The authors gave the same response as above.)

Round 2

Reviewer 1 Report

OK. I'm satisfied with the changes that you have made in the revised version. Thanks

Reviewer 4 Report

Appreciate the efforts of authors.

Remove 2 Methodology it should be “ 2 Theoretical Foundation